# DUSK: Do Not Unlearn Shared Knowledge

## Abstract

Machine unlearning aims to remove "forget" data while preserving knowledge from the "retain" data, but what should happen when they share content? According to the formal definition of machine unlearning, an unlearned model should be indistinguishable from a retrained model trained solely on the retain set. This means that shared knowledge must remain while unique forget content is removed. We introduce DUSK, the first benchmark to evaluate unlearning under realistic knowledge overlap. DUSK constructs documents that contain shared and unique knowledge across five different styles and defines seven metrics to test whether methods erase forget-specific expressions without discarding valid shared facts. By evaluating nine recent approaches, we find that despite the frequent removal of surface text, current methods struggle to distinguish unique from shared knowledge, either removing shared knowledge that should be preserved or failing to erase forget-specific information. DUSK provides a controlled, reproducible testbed for diagnosing such failures and guiding more precise unlearning algorithms.

## 1 Introduction

Large language models (LLMs) are typically trained on web-scale corpora that include copyrighted materials, personal data, and user-generated content (Carlini et al., 2021; Nasr et al., 2025). As these models are deployed in real-world applications, individuals and organizations increasingly demand the removal of specific training examples due to legal and ethical concerns. These demands are driven by privacy regulations such as the GDPR (Voigt & Von dem Bussche, 2017) and reinforced by recent lawsuits (Grynbaum & Mac, 2023; *Tremblay v. OpenAI, Inc.*, 2023; *DOE 1 v. GitHub, Inc.*, 2022) over the unauthorized use of proprietary content. This has led to growing interest in machine unlearning (Nguyen et al., 2025; Liu et al., 2025), which focuses on removing the influence of *forget data* (e.g., copyrighted documents) from a trained model without retraining from scratch, as preserving information from *retain data*. To evaluate unlearning algorithms, several benchmarks have recently been proposed (Maini et al., 2024; Shi et al., 2025; Jin et al., 2024). For example, MUSE (Shi et al., 2025) targets copyright-related scenarios by focusing on the removal of entire documents. In such cases, unlearning algorithms are expected to erase both verbatim text and the underlying knowledge from the forget set, while preserving information from the retain set.

However, existing unlearning benchmarks often assume that the forget set and retain set are disjoint, overlooking the complexity of real-world data where documents frequently share information. For example, if a New York Times article is subject to a deletion request, it may describe, "A 6.2 magnitude earthquake struck Tokyo on Monday," while a Wikipedia article in the retain set states, "A strong tremor shook the Japanese capital at the start of the week." Despite stylistic differences, both convey the same factual content. This raises a key question for unlearning: *what should a model do when forget and retain sets overlap?* Intuitively, the model should preserve any information that exists in the retain set, since it can be learned from the retain set alone without access to the forget set. This also aligns with the definition of machine unlearning, which requires the unlearned model to be indistinguishable from one retrained solely on the retain set.

To address this gap, we introduce DUSK, a benchmark for evaluating unlearning in settings where forget and retain sets share overlapping knowledge. DUSK constructs document sets that describe the same factual content in different styles, allowing clear attribution: some information is unique to the forget set, while other content remains supported by the retain set. The dataset consists of

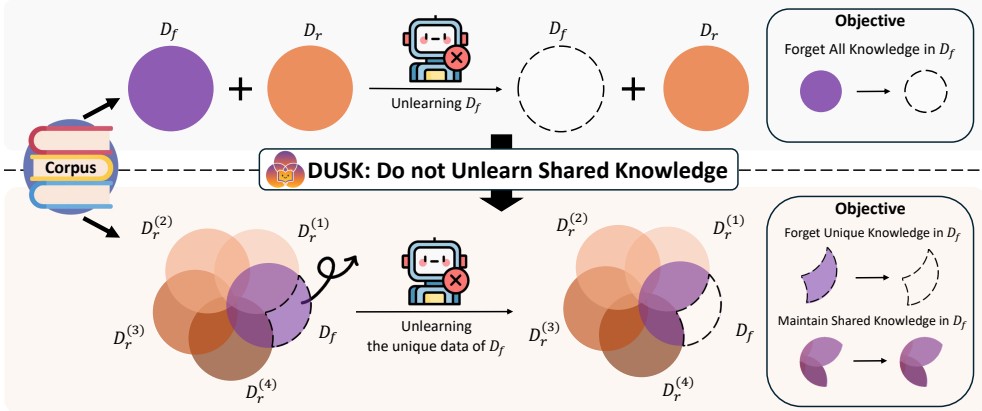

**Figure 1:** 🧑‍🔬 **DUSK provides a realistic unlearning evaluation scenario where forget documents ($\mathcal{D}_f$) contain both unique information to be forgotten and shared knowledge that must be preserved.** Unlike conventional setups that naively erase entire forget sets, DUSK evaluates whether unlearning methods can selectively remove sensitive information while retaining shared knowledge supported by other documents in the retain set ($\mathcal{D}_r$), which is not subject to forgetting.

120 synthetic professor profiles organized into five documents, each containing unique and shared profiles. We construct DUSK with synthetic data to enable precise control and systematic evaluation of unlearning outcomes, as overlaps between forget and retain information are often ambiguous in real-world data, yet essential for meaningful evaluation. This choice follows prior work (Maini et al., 2024; Yoon et al., 2025), which similarly uses synthetic setups for controlled unlearning analysis. In this way, DUSK offers a simple yet effective benchmark that, for the first time, directly evaluates unlearning under shared knowledge conditions. An overview is shown in Figure 1.

DUSK defines seven evaluation metrics: (1) Verbatim Memorization, which checks whether exact text from the forget set has been fully removed; (2) Unique Forget Knowledge, which measures whether forget-only content is effectively erased; (3) Shared Knowledge, which ensures that overlapping information is preserved; (4) Unique Retain Knowledge, which verifies that retain-only facts remain intact; (5) Downstream Capability, which evaluates whether the model maintains general utility after unlearning; (6) Privacy Leakage, which inspects for residual leakage of forget set information; and (7) Retain Deviation, which assesses whether the model's behavior on the retain set is preserved.

We evaluate nine unlearning methods on the DUSK benchmark. Our results reveal a critical limitation of existing methods: while these methods aim to remove content uniquely attributable to the forget set, they often struggle to fully separate this content from information still supported by the retain set. This incomplete disentanglement can degrade shared knowledge, compromising the model's overall utility. These findings highlight a fundamental challenge in current unlearning approaches: precisely distinguishing forget-specific information from retained knowledge. To support further research, we will release DUSK as a public benchmark for evaluating unlearning in real-world scenarios.

## 2 RELATED WORK

**Machine Unlearning in LLMs: Methods and Applications.** Machine unlearning aims to selectively remove the influence of *forget data* from a trained model while preserving its performance on *retain data* (Cao & Yang, 2015; Brophy & Lowd, 2021; Jeon et al., 2024). Recent efforts have extended unlearning techniques to large language models (LLMs) (Liu et al., 2025), enabling their use in a range of applications such as removing copyrighted content (Kassem et al., 2023; Wei et al., 2024), eliminating sensitive or harmful knowledge (Maini et al., 2024; Yousefpour et al., 2025; Zhang et al., 2024b), mitigating bias (Dige et al., 2024; Jeung et al., 2025a), and performing model editing (Guo et al., 2025). Most methods achieve unlearning by fine-tuning on the forget data (Chen & Yang, 2023; Jia et al., 2024; Cao & Yang, 2015; Barbulescu & Triantafillou, 2024), commonly using gradient ascent (Jang et al., 2023) or preference optimization (Zhang et al., 2024a). To scale these methods to large models, recent work has explored approaches such as guardrail-based techniques (Thaker et al., 2024; Gao et al., 2025), and in-context unlearning (Pawelczyk et al., 2024). Despite this progress,

recent studies have highlighted the fragility of current unlearning techniques (Hu et al., 2025; Lynch et al., 2024; Thaker et al., 2025; Zhang et al., 2025b; Joshi et al., 2024; Jeung et al., 2025b), revealing the fundamental challenges in achieving robust and reliable unlearning in practice.

**Machine Unlearning in LLMs: Benchmarks.** As machine unlearning methods for LLMs evolve, the need for comprehensive evaluation benchmarks has become increasingly important. Early work introduced the "Who is Harry Potter" (WHP) task (Eldan & Russinovich, 2023), which targets entity-specific forgetting by fine-tuning models on fictional corpora and evaluating unlearning through related prompts while monitoring retention on unrelated tasks. Subsequent efforts expand the scope to hazardous knowledge, with the WMDP (Li et al., 2024) focusing on removing information related to biosecurity and cybersecurity without compromising general model capabilities. To enable controlled evaluation of unlearning, TOFU (Maini et al., 2024) constructs synthetic author profiles with associated question-answer pairs generated by GPT-4. This synthetic setup ensures that the model's knowledge of these authors originates solely from the fine-tuning process, allowing for precise assessment of unlearning effectiveness. MUSE (Shi et al., 2025) introduces a six-dimensional evaluation framework for unlearning algorithms, spanning forgetting effectiveness, privacy leakage, utility retention, scalability, and sustainability. It aims to remove both verbatim content and underlying knowledge within the forget set. CoTAEval (Wei et al., 2024) instead focuses on a narrower goal, removing only verbatim memorization while explicitly preserving the associated knowledge. Building on this line of work, RWKU (Jin et al., 2024) proposes a more challenging setting where neither the forget nor retain corpus is accessible. It targets the removal of widely known real-world knowledge, such as facts about 200 famous individuals , and evaluates performance via membership inference attacks, adversarial probes, and tasks assessing reasoning, truthfulness, and fluency. However, existing benchmarks tend to assume that the forget set contains only information to be removed, overlooking the realistic scenario where forget documents often contain both information that should be forgotten and information that should be retained. We address this gap by introducing DUSK, a benchmark for multi-source unlearning where forget-specific and retained knowledge coexist within each document.

# 3 THE 🔮 DUSK BENCHMARK

## 3.1 PROBLEM SETTING

The central principle of traditional machine unlearning is that the unlearned model should be essentially indistinguishable from a model retrained from scratch on the retain set alone. Because such a retrained model naturally preserves all knowledge supported by the retain set, **shared knowledge must strictly remain**, while only the content unique to the forget set should be removed.

DUSK is designed to evaluate this requirement under the realistic conditions where the forget and retain sets may frequently overlap in practice. Each document therefore contains the following two types of content: (1) **shared knowledge**, factual information that appears in the multiple documents and should remain accessible even if one document is deleted, and (2) **unique knowledge**, information specific to a single document that should be forgotten when that document is removed.

Given a forget request for a particular document, we assess whether an unlearning algorithm can:

1. Preserve shared knowledge supported by the retain set,
2. Remove unique knowledge from the forgotten document, and
3. Preserve unique knowledge from other (non-forgotten) documents.

This task formulation highlights the inherent practical difficulty of unlearning in the multi-source environments: the boundary between forgetting and retaining is often blurred, yet effective methods must closely approximate the outcomes of a model retrained on the retain set.

## 3.2 PROBLEM FORMULATION AND NOTATIONS

Let $f_\theta$ be a target model trained on a dataset $\mathcal{D}$, and let $\mathcal{D}_f \subset \mathcal{D}$ denote the subset of training data targeted for removal (i.e., forget set[1]). The goal is to produce an unlearned model $f_{\theta'}$ that no

---

[1]DUSK focuses on document-level unlearning, reflecting real-world cases such as *The New York Times v. Microsoft*, where deletion requests specify complete documents, making the forget set clearly defined.

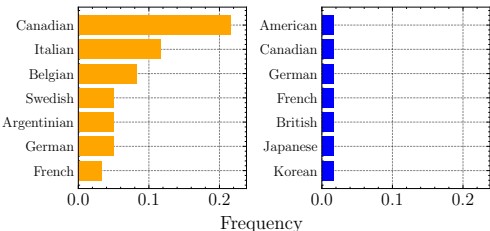 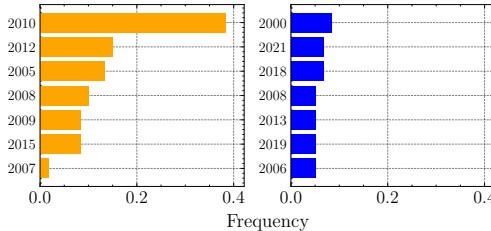

**Figure 2: Distributions of country of nationality (left) and graduate year (right) for the seven most common attributes in GPT-4 outputs.** This reveals mode collapse with default prompts, disproportionately favoring frequent values like "Canada" and "2010." After prompt refinement, distributions become more balanced, reflecting a more diverse attribute range.

longer exposes the information contained in $\mathcal{D}_f$, while maintaining utility on the remaining data, $\mathcal{D}_r = \mathcal{D} \setminus \mathcal{D}_f$. Ideally, $f_{\theta'}$ should be indistinguishable from a model retrained from scratch on $\mathcal{D}_r$.

We define $\mathcal{K}_f$ as the knowledge contained in the forget set $\mathcal{D}_f$, and $\mathcal{K}_r$ as the knowledge contained in the retain set $\mathcal{D}_r$. Prior works often assume that $\mathcal{K}_f \cap \mathcal{K}_r = \varnothing$, meaning the knowledge from the forget and retain sets does not overlap. However, this assumption rarely holds in practice. In many real-world cases, the same information appears across both sets in different phrasings or styles. As a result, an effective unlearning method must identify and remove only the portion of knowledge that is uniquely attributable to $\mathcal{D}_f$, while preserving content that is also supported by $\mathcal{D}_r$.

### 3.3 THE 🔮 DUSK DATASET CONSTRUCTION

To ensure precise control over the origin and overlap of information, we construct a synthetic dataset inspired by TOFU (Maini et al., 2024). It comprises 120 fictitious professor profiles, each built using structured attributes, such as academic department and institutional affiliation. Because these profiles are entirely absent from any pretrained corpus, they guarantee a clean experimental environment with clearly defined forget and retain sets. We generate five documents, each representing an independent data source. These documents collectively cover all 120 profiles, including both shared profiles that appear in multiple documents and unique profiles that are present in only one, enabling fine-grained control over content attribution across sources. The profiles are partitioned into two disjoint subsets:

- **Shared Knowledge**: 60 profiles appear in all five documents, each presented in a different style. These profiles represent redundantly supported knowledge that should be preserved regardless of which document is unlearned.
- **Unique Knowledge**: The remaining 60 profiles are evenly distributed across the five documents, with each document containing 12 unique profiles that do not appear in any other. These profiles represent document-specific information that should be forgotten when the corresponding document is unlearned.

We also create a holdout set ($\mathcal{D}_h$) consisting of 120 professors that do not overlap with $\mathcal{D}_r$ or $\mathcal{D}_f$. In constructing, we follow the same process used to construct $\mathcal{D}_r$ and $\mathcal{D}_f$. The holdout set has never been included in the training data for either the Retrain model or the Target model.

### 3.3.1 DATA GENERATION PIPELINE

**Knowledge Source.** We begin by generating a knowledge base of 120 fictitious professor profiles, each represented by 20 question–answer (QA) pairs covering attributes such as birth year, nationality, and academic history. The QA pairs are synthesized with GPT-4 to ensure fluency and diversity.

However, we observe that GPT-4 exhibits strong biases in generating attribute values. For example, as shown in Figure 2, GPT-4 disproportionately favors the year *2010* for graduation years, with this value appearing in nearly 40% of cases. Similarly, nationality values are skewed, with a strong preference for *Canadian*, which in turn affects related fields like birthplace and affiliated universities.

To mitigate these biases, we iteratively refine the prompt design. When we observe skewed distributions in attributes in the generated data, we adjust the prompts accordingly by explicitly specifying

underrepresented categories or by sampling values like graduation year uniformly within a reasonable range and then regenerate the data. Through four iterations of prompt-based refinement, we obtain more balanced and realistic outputs from GPT-4. As illustrated in Figure 2, the final set of profiles exhibits significantly improved attribute diversity compared to the initial generations. Additional implementation details, along with privacy and integrity audits, are described in Appendix A.1.

**Document Construction.**   Using the processed QA profiles as source knowledge, we construct five distinct documents, denoted as $\{\mathcal{D}_i\}_{i=1}^5$, each expressing the underlying content through a different narrative style. We designate $\mathcal{D}_1$ as the forget set and define the retain set as the union of the remaining documents, $\mathcal{D}_r = \bigcup_{i=2}^5 \mathcal{D}_i$. To simulate stylistic diversity reflective of real-world corpora, we format each document using a distinct narrative genre: **Chronological**, which presents profiles as career timelines ordered by milestones; **Feature Story**, which uses narrative-driven descriptions akin to editorial articles; **Interview**, which formats profiles as fictional Q&A sessions with conversational tone; **Inverted Pyramid**, which follows journalistic convention by placing key facts first; and **Listicle**, which presents profiles in ranked or grouped lists using bullet-point highlights.

Each document is composed of the same 60 shared profiles and 12 unique profiles in style-specific templates. This setup allows us to evaluate whether unlearning methods can selectively remove isolated information while preserving general knowledge under stylistic or structural variation. Corresponding text examples for the shared knowledge in each document style are provided in Appendix A.2.

### 3.4   The 🧠 DUSK Evaluation

The DUSK evaluation framework characterizes unlearning behavior in three dimensions: (1) *what should be forgotten*, (2) *what should be retained*, and (3) *whether the model behaves as if trained only on the retain set*. An effective unlearning method should eliminate not only verbatim content from the forget set but also knowledge uniquely attributable to it. It should retain shared and exclusive information in the retain set and preserve downstream capabilities, while ensuring the model's behavior becomes indistinguishable from that of a model trained without access to the forget set.

#### 3.4.1   Forget Assessment

**Verbatim Memorization (VM).**   We assess whether the unlearned model can still reproduce exact phrasings from the forget set, even when the underlying knowledge is shared across both forget and retain sets. While such shared knowledge should be preserved, any specific wording originating from the forget document must be removed. This is particularly critical because the forget set often contains copyright-protected material and regenerating such text would indicate incomplete unlearning.

To comprehensively evaluate memorization, we prompt the model with partial sequences from $\mathcal{D}_f$, denoted as $d_{[:\ell]}$, and compare the model's continuations to the original text $d_{[\ell+1:]}$ across multiple metrics. Specifically, we use ROUGE-1 and ROUGE-L (F1 scores) to measure overall lexical and structural overlap, and their Recall variants to emphasize ground-truth coverage. We further include the Levenshtein Distance (Levenshtein et al., 1966) to quantify the minimum number of edits required for alignment, Longest Common Subsequence (LCS) for sequential token overlap, and Cosine Similarity (Cer et al., 2017) for embedding-level semantic similarity. Here, higher scores indicate that the model remains capable of reproducing text that should have been forgotten.

**Unique Forget Knowledge (UFK).**   We evaluate whether the model retains knowledge $\mathcal{K}_f \smallsetminus \mathcal{K}_r$ that is uniquely attributable to the forget set $\mathcal{D}_f$ by prompting it with targeted questions. Overlap between the model's responses and the correct answers is measured using ROUGE-L scores (Lin, 2004), where lower scores indicate more effective unlearning of forget source-specific information.

#### 3.4.2   Retain Assessment

**Shared Knowledge (SK).**   Unlike prior benchmarks that aim to remove all knowledge about the forget set, multi-source scenarios, where training data originates from diverse and overlapping sources, often involve shared knowledge appearing in both $\mathcal{D}_f$ and $\mathcal{D}_r$. In such cases, indiscriminately unlearning the entire forget set risks discarding overlapping content that should remain accessible.

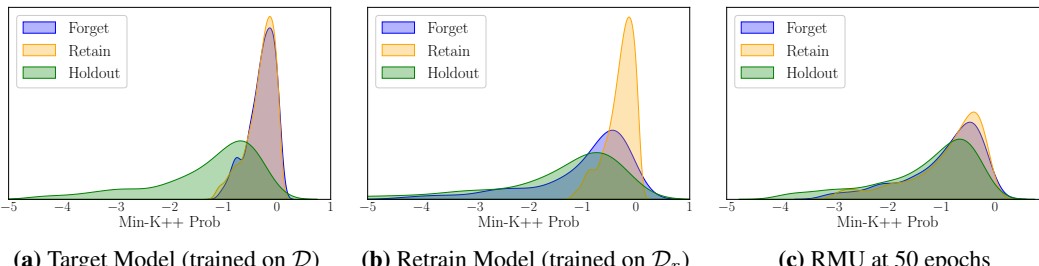

**(a)** Target Model (trained on $\mathcal{D}$)     **(b)** Retrain Model (trained on $\mathcal{D}_r$)     **(c)** RMU at 50 epochs

**Figure 3: Min-K++ Probability Distributions over $\mathcal{D}_f$, $\mathcal{D}_r$, and $\mathcal{D}_h$.** (a) Target model trained on both $\mathcal{D}_f$ and $\mathcal{D}_r$ shows higher probabilities, reflecting retained knowledge, while $\mathcal{D}_h$ remains lower. (b) Retrain model reduces probabilities on $\mathcal{D}_f$, as they are not trained on it, representing ideal unlearning. (c) Some unlearned models achieve ideal low probabilities on $\mathcal{D}_f$ but risk collapsing $\mathcal{D}_r$, as detected by Retain Deviation.

To assess the preservation of shared knowledge, we construct queries targeting $\mathcal{K}_f \cap \mathcal{K}_r$ (i.e., information present in both the forget and retain sets) and evaluate the model's responses using ROUGE-L scores measured against the ground truth answers. High scores indicate successful preservation, while low scores indicate unintended forgetting caused by overly aggressive unlearning.

**Unique Retain Knowledge (URK).** To assess the preservation of retain-exclusive knowledge $\mathcal{K}_r \setminus \mathcal{K}_f$, we specifically construct queries answerable only from $\mathcal{D}_r$ and not from $\mathcal{D}_f$. Model responses are then compared against ground truth answers using ROUGE-L scores, where higher scores indicate successful retention without the unintended removal of unique retain content.

**Downstream Capability (DC).** We verify that the model's fundamental capabilities, such as reasoning, factual consistency, and fairness are indeed preserved after unlearning. We assess performance across six downstream tasks: MMLU for broad knowledge and ability (Hendrycks et al., 2021), ARC-c for challenging reasoning (Clark et al., 2018), GSM8K for arithmetic and problem-solving (Cobbe et al., 2021), TriviaQA for factual recall (Joshi et al., 2017), TruthfulQA (MC1) for truthfulness evaluation (Lin et al., 2022), and BBQ for social bias probing under ambiguity (Parrish et al., 2022). A successful unlearning method should remove only the targeted information while maintaining the core competencies and showing strong performance on downstream tasks.

### 3.4.3 DISTRIBUTIONAL ASSESSMENT

**Privacy Leakage.** We assess privacy leakage by evaluating whether any behavioral traces from the forget set remain in the unlearned model. Following the MUSE benchmark (Shi et al., 2025), we adopt a membership inference attack (MIA) framework and apply Min-K%++ (Zhang et al., 2025a) to capture subtle distributional differences. Specifically, as shown in Figure 3, we measure the model's ability to distinguish samples from forget set ($\mathcal{D}_f$) and a holdout set ($\mathcal{D}_h$), which consists of unseen data. We report the AUC-ROC of this discrimination task and normalize it relative to a Retrain model that excludes $\mathcal{D}_f$ for training. Privacy Leakage score is defined as:

$$\text{PrivacyLeak} := \frac{\text{AUC}_{\text{unlearn}}(\mathcal{D}_f, \mathcal{D}_h) - \text{AUC}_{\text{retrain}}(\mathcal{D}_f, \mathcal{D}_h)}{\text{AUC}_{\text{retrain}}(\mathcal{D}_f, \mathcal{D}_h)}.$$

A Privacy Leakage value close to zero indicates that the unlearned model treats $\mathcal{D}_f$ similarly to $\mathcal{D}_h$, suggesting successful unlearning of $\mathcal{D}_f$. Values below zero indicate under-unlearning, where the model continues to assign high probability to forget data. Conversely, values above zero reflect over-unlearning, where the model suppresses forget set too aggressively, leading to excessive forgetting. As shown in Figure 3b, it is important to note that $\mathcal{D}_f$ and $\mathcal{D}_h$ are not expected to follow identical distributions even after ideal unlearning. This is because $\mathcal{D}_f$ may contain shared knowledge that overlaps with the $\mathcal{D}_r$, while $\mathcal{D}_h$ consists entirely of unseen content.

**Retain Deviation.** Unlearning often disrupts the model's ability to distinguish $\mathcal{D}_r$, causing both $\mathcal{D}_f$ and $\mathcal{D}_r$ to collapse toward $\mathcal{D}_h$, as illustrated in Figure 3c. This issue becomes more pronounced in multi-source unlearning scenarios, where overlapping information between $\mathcal{D}_f$ and $\mathcal{D}_r$ makes $\mathcal{D}_r$

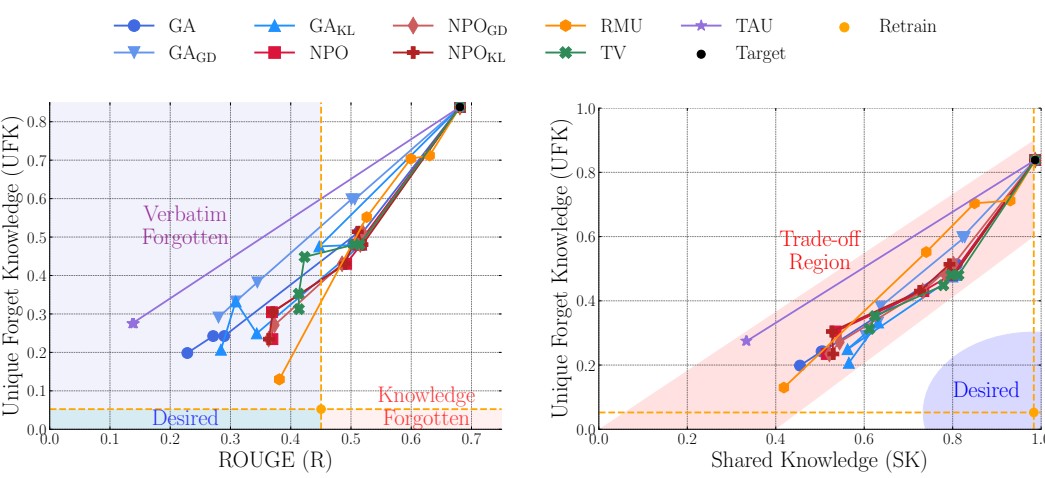

(a) **Forget Verbatim vs. Forget Knowledge.**    (b) **Forget knowledge vs. Shared knowledge.**

Figure 4: **Two-dimensional analysis of unlearning dynamics.** We visualize model trajectories over multiple epochs to illustrate key trade-offs in DUSK. (a) shows the trade-off between verbatim and knowledge forgetting, while (b) shows the trade-off between shared knowledge and unique forget knowledge.

more vulnerable to unintended forgetting. To quantify this side effect, we introduce a supplementary metric, Retain Deviation, which applies the same MIA framework to $\mathcal{D}_r$ and is defined as:

$$\text{RetainDeviation} := \left| \frac{\text{AUC}_{\text{unlearn}}(\mathcal{D}_r, \mathcal{D}_h) - \text{AUC}_{\text{retrain}}(\mathcal{D}_r, \mathcal{D}_h)}{\text{AUC}_{\text{retrain}}(\mathcal{D}_r, \mathcal{D}_h)} \right|.$$

A low Retain Deviation score indicates that the model retains its original capabilities on $\mathcal{D}_r$ after unlearning. As Retain Deviation increases, the model's behavior on $\mathcal{D}_r$ diverges from its original state, suggesting that important retained knowledge may not have been properly preserved.

## 4 EXPERIMENTS

### 4.1 UNLEARNING METHODS

**Removing Forget Set.** We introduce five unlearning methods designed to effectively remove the influence of forget data. Gradient Ascent (GA) (Jang et al., 2023) maximizes the loss on the forget set $\mathcal{D}_f$, reducing the model's ability to reproduce its content. Negative Preference Optimization (NPO) (Zhang et al., 2024a) extends Direct Preference Optimization (DPO) (Rafailov et al., 2023) for unlearning by treating samples in $\mathcal{D}_f$ as negative preferences relative to a Target model. Representation Misdirection for Unlearning (RMU) (Li et al., 2024) modifies intermediate representations by pushing activations of the forget set toward random directions, while aligning retain set activations with those of a frozen Target model. Task Vector (TV) (Ilharco et al., 2023) removes the influence of $\mathcal{D}_f$ by computing the parameter changes caused by fine-tuning on $\mathcal{D}_f$ and subtracting them from the original model weights. Lastly, Task Arithmetic for Unlearning (TAU) (Barbulescu & Triantafillou, 2024) performs two steps: it first applies gradient ascent selectively to samples with high memorization scores, and then conducts task vector subtraction as described above.

**Preserving Retain Set.** To maintain model utility during unlearning, we incorporate two regularization losses. Gradient Descent (GD) preserves performance on the retain set $\mathcal{D}_r$ by applying prediction loss. This ensures that removing $\mathcal{D}_f$ does not excessively degrade the model's behavior on unrelated data. KL Divergence (KL) (Hinton et al., 2014) encourages consistency between the unlearned model's predictions on $\mathcal{D}_r$ and those of a Target model. By minimizing this divergence, the model retains useful information and softly constraining deviation from its original output distribution.

Consequently, we evaluate nine total configurations: GA, GA$_{GD}$, GA$_{KL}$, NPO, NPO$_{GD}$, NPO$_{KL}$, RMU, TV, and TAU, where the suffix indicates an added utility-preserving objective. Additional details about these methods can be found in Appendix B.1.

**Table 1: Impact of Unlearning on UFK, SK, URK, and DC.** We report both raw values and their differences relative to the Retrain model. Red indicates higher values, and Blue indicates lower values, with darker shades indicating greater magnitude. DC is calculated by averaging of six benchmarks explained in Section 3.4.2.

| | Unique Forget Knowledge UFK (↓) | | Shared Knowledge SK (↑) | | Unique Retain Knowledge URK (↑) | | Downstream Capability DC (↑) | |
|---|---|---|---|---|---|---|---|---|
| Target | 83.8 | | 98.6 | | 88.7 | | 40.3 | |
| Retrain | **5.2** | | **98.3** | | **84.8** | | **40.6** | |
| GA | 24.3 | (+367%) | 50.7 | (−48%) | 52.5 | (−38%) | 37.5 | (−8%) |
| GA$_{GD}$ | 38.2 | (+635%) | 63.7 | (−35%) | 63.6 | (−25%) | 39.0 | (−4%) |
| GA$_{KL}$ | 24.9 | (+379%) | 56.2 | (−43%) | 57.4 | (−32%) | 38.6 | (−5%) |
| NPO | 43.0 | (+727%) | 73.4 | (−25%) | 69.6 | (−18%) | 39.5 | (−3%) |
| NPO$_{GD}$ | 27.1 | (+421%) | 54.4 | (−45%) | 51.1 | (−40%) | 37.8 | (−7%) |
| NPO$_{KL}$ | 30.4 | (+485%) | 52.7 | (−46%) | 52.8 | (−38%) | 37.7 | (−7%) |
| RMU | 55.1 | (+960%) | 74.0 | (−25%) | 64.0 | (−25%) | 39.1 | (−4%) |
| TV | 35.3 | (+579%) | 62.4 | (−37%) | 69.1 | (−19%) | 40.3 | (−1%) |
| TAU | 27.5 | (+429%) | 33.5 | (−66%) | 50.7 | (−40%) | 35.8 | (−12%) |

## 4.2 EXPERIMENTAL SETUP

We begin with a pretrained base model (`LLaMA-3-8B` (Dubey et al., 2024)). Target model is obtained by fine-tuning this base model on the full corpus ($\mathcal{D}_r \cup \mathcal{D}_f$) for 5 epochs with a learning rate of $1 \times 10^{-5}$, following prior benchmarks (Maini et al., 2024; Shi et al., 2025). Retrain model is trained solely on the retain set $\mathcal{D}_r$ under the same setup.

For all unlearning methods, we adopt the AdamW optimizer with a learning rate of $1 \times 10^{-5}$ and a batch size of 32, using the first epoch as a warm-up phase, consistent with prior work (Maini et al., 2024). Since unlearning performance is sensitive to the number of training epochs, we standardize the stopping criterion across methods: We terminate unlearning at the first epoch where the Unique Retain Knowledge (URK) score falls below 70. This ensures comparable utility levels, enabling fair and consistent comparisons across methods. Further implementation details are provided in Appendix B.

## 4.3 UNLEARNING RESULTS

### 4.3.1 FORGET ASSESSMENT RESULTS.

**Verbatim Memorization (VM).** We first evaluate whether unlearning methods can suppress verbatim memorization of the forget set. Given prefix excerpts from $\mathcal{D}_f$, we prompt the model to continue the text and measure similarity to the original using ROUGE, Levenshtein Distance, LCS, and Cosine Similarity. As shown in Figure 4a, most methods reduce lexical overlap with the original text, with TAU consistently achieving the largest reductions, indicating strong suppression of verbatim memorization. Full results for verbatim memorization are provided in Appendix C.1.

**Unique Forget Knowledge (UFK).** To assess whether unlearning removes not just surface expressions but deeper factual knowledge, we evaluate models on Unique Forget Knowledge (UFK), consisting of questions relying exclusively on information from $\mathcal{D}_f$. As shown in Figure 4a, plotting average ROUGE scores against UFK accuracy, most methods shift into the Verbatim Forgotten region, showing effective surface-level suppression. Yet, they largely fail to erase underlying facts, as models continue to answer UFK questions correctly, implying that knowledge forgetting remains incomplete.

### 4.3.2 RETAIN ASSESSMENT RESULTS.

**Shared Knowledge (SK) and Unique Retain Knowledge (URK).** We evaluate whether unlearning unintentionally erases information that should be preserved. As shown in Figure 4b, almost all contemporary unlearning methods not only reduce UFK scores as intended but also substantially degrade SK scores, revealing a consistent failure to preserve shared knowledge. This suggests that existing unlearning methods tend to degrade the utility of the model by also removing shared

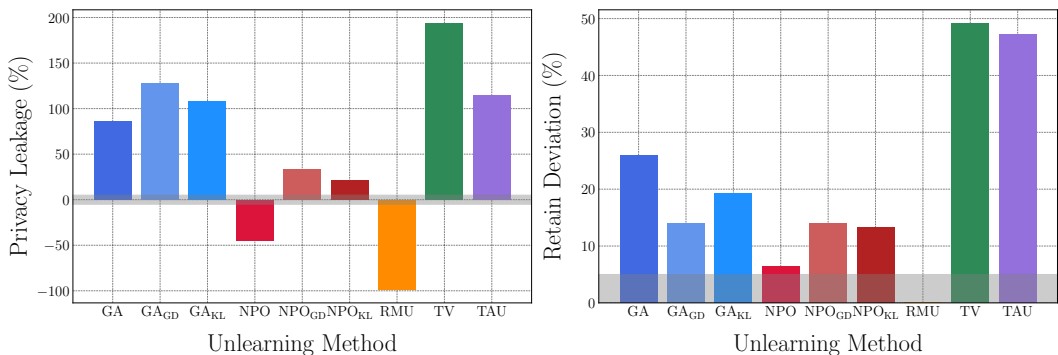

Figure 5: **Privacy Leakage and Retain Deviation Analysis.** Gray bands indicate optimal bounds: $[-5\%, 5\%]$ for leakage and $[0\%, 5\%]$ for deviation. Values outside these ranges reflect under-unlearning (below $-5\%$), over-unlearning (above $5\%$) in leakage, or degradation of retained knowledge (above $5\%$) in deviation.

knowledge that overlaps with the forget set. Furthermore, as highlighted in Table 1, SK suffers greater accuracy degradation than URK across most methods. Since SK spans both forget and retain sets, unlearning that targets the forget set inadvertently harms overlapping knowledge that should ideally be preserved. These findings reveal a key limitation of current approaches, as they struggle to selectively unlearn knowledge associated with the forget set without also disrupting shared knowledge.

**Downstream Capability (DC).** We evaluate general capability after unlearning using a range of downstream tasks, including MMLU, ARC-c, GSM8K, TriviaQA, TruthfulQA, and BBQ. Across most methods, performance on these tasks remains stable, with only modest declines relative to the Retrain model, indicating that core capabilities such as reasoning, factual recall, and fairness are largely preserved. By contrast, as shown in Table 1, knowledge closely related to the forget set, captured by SK and URK, degrades far more than the overall downstream capability (see Appendix Table 7). Taken together, this pattern suggests that even when broad capabilities are maintained, knowledge conceptually adjacent to the forget set remains highly vulnerable to collateral forgetting. Full results for downstream capability are presented in Appendix C.2.

### 4.3.3 DISTRIBUTIONAL ASSESSMENT RESULTS.

**Privacy Leakage and Retain Deviation.** Successful unlearning is ideally indicated by both the Privacy Leakage and Retain Deviation values close to zero. However, we observe two particularly representative patterns in their joint behavior that fall short of this ideal as illustrated in Figure 5. Most cases exhibit over-unlearning along with rising Retain Deviation, where unlearning $\mathcal{D}_f$ leads to unintended changes in the model's responses to $\mathcal{D}_r$ due to shared knowledge. In contrast, NPO exhibits under-unlearning, yet still show a rising Retain Deviation. This suggests that even before $\mathcal{D}_f$ is fully unlearned, the model's performance on $\mathcal{D}_r$ can already deteriorate due to entangled representations arising from overlapping knowledge. Taken together, these findings indicate that under realistic conditions where $\mathcal{D}_f$ and $\mathcal{D}_r$ are not disjoint, no method can completely remove the influence of $\mathcal{D}_f$ while fully preserving the model's behavior on $\mathcal{D}_r$. This underscores the importance of jointly monitoring Privacy Leakage and Retain Deviation in multi-source unlearning scenarios.

## 5 CONCLUSION

We introduce 🤖 DUSK, a benchmark for evaluating machine unlearning in realistic multi-source scenarios, where forget data often overlaps with retain data. Unlike prior work, DUSK explicitly separates unique and shared knowledge, providing a fine-grained testbed for assessing unlearning performance. Our experiments reveal that while most existing unlearning methods effectively remove verbatim content, they often fail to disentangle forget-specific knowledge from overlapping facts. This failure leads to unintended degradation of both shared and retain-only knowledge that should have been preserved. We hope DUSK will serve as a foundation for advancing more precise and reliable unlearning methods, bridging the gap between theoretical formulations and real-world applications.

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

# A  DETAILS OF 🦝 DUSK

## A.1  DATASET CONSTRUCTION DETAILS

**Table 2:** Professor Information Fields

| # | Field | Description |
|---|-------|-------------|
| 1 | Nationality | The professor's nationality. |
| 2 | Born | The birthplace of the professor. |
| 3 | Closest Colleague | The professor's closest colleague or collaborator. |
| 4 | Year of birth | The birth year of the professor. |
| 5 | Department | The major of the professor is affiliated with. |
| 6 | Award | The most prestigious award received by the professor. |
| 7 | School | The fictitious university where the professor teaches. |
| 8 | Best paper | The most well-known and fictitious research paper authored by the professor. |
| 9 | Office number | The room number where the professor's office is located. |
| 10 | E-mail | A fictitious email address associated with the professor. |
| 11 | Research Interests | The professor's main research areas. |
| 12 | Funded Projects | Major fictitious research projects funded under the professor's name. |
| 13 | Patents | Any fictitious patents held by the professor. |
| 14 | Course | The fictitious course(s) taught by the professor. |
| 15 | Hobby | The professor's main hobby outside of work. |
| 16 | Alma Mater | The university where the professor received their PhD. |
| 17 | Favorite Theorem | The professor's favorite theorem or concept. |
| 18 | Religion | The professor's religious affiliation. |
| 19 | Lab name | The fictitious name of the professor's laboratory. |
| 20 | Year of employment | The year the professor was appointed to their current university. |

**Knowledge Source.**  To generate a dataset of 120 fictional professors, we use GPT-4 to produce 20 question–answer pairs for each individual, resulting in a total of 2,400 QA pairs. Types of questions used for each professor are listed in Table 2, covering a wide range of biographical, academic, and professional attributes to ensure diversity and richness in the generated data.

To further improve representational balance, we refine the prompts used during generation by controlling several key attributes. For *country of nationality*, we manually select 60 distinct countries, which naturally increases diversity in *birthplace* as well, since GPT-4 tends to produce regionally coherent outputs. For *religion*, we choose eight widely practiced belief systems—Christian, Muslim, Jewish, Hindu, Buddhist, Agnostic, Atheist, and Spiritual—and assign them uniformly across the dataset. For temporal attributes such as *year of birth* and *year of employment*, which otherwise show skewed distributions, we sample values uniformly within a reasonable range and include them directly in the prompt. The effectiveness of prompt refinement is reflected in the attribute distributions shown in Figure 6 and Figure 7. Compared to the initial outputs, which display strong mode collapse in attributes such as nationality and employment year, the refined versions demonstrate significantly more balanced and diverse distributions. Figure 8 shows the final prompt we used for QA generation.

After generating the full QA sets, we perform a final validation step to identify any duplicate professor names. This ensures the dataset can support a realistic and rigorous unlearning scenario, where identifying and selectively removing information about specific individuals is required.

**Dataset Construction.**  For each professor, we create profiles based on information generated from QA pairs with prompt in Figure 9. Each professor's information is used to create five profiles in five different styles: Chronological, Feature Story, Interview, Inverted Pyramid, and Listicle, resulting in a total of 600 professor profiles (120 per style). These profiles are divided into shared knowledge and unique knowledge components.

The shared knowledge set consists of 60 professors, each represented by a single profile in each style, resulting in 300 profiles (60 professors × 5 styles). These 60 professors are included in all five style-specific documents, with each document containing the same set of 60 professors, but with their profiles presented in different styles.

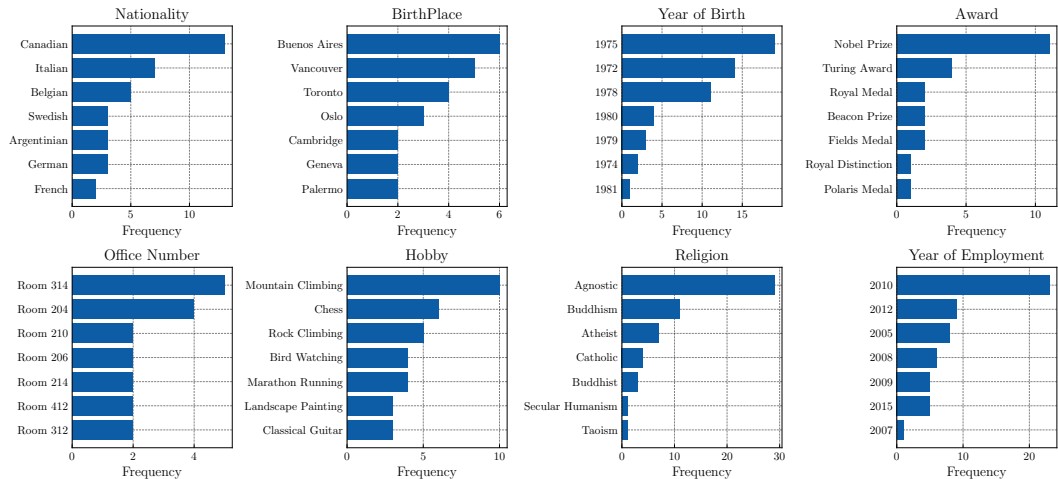

Figure 6: **Distributions of seven most common attributes in GPT-4 outputs before prompt refinement.** Several features exhibit mode collapse, with overrepresentation of specific values such as "Canadian" for nationality, "2010" for year of employment, and "Agnostic" for religion, reflecting bias in uncontrolled generation.

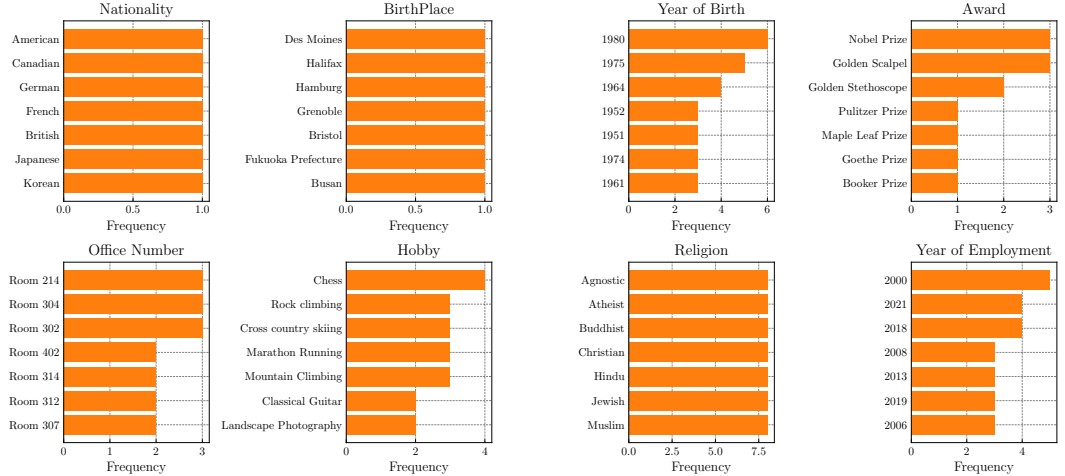

Figure 7: **Distributions of seven most common attributes after prompt refinement.** The frequency of values across attributes such as nationality, religion, and year of employment is more balanced, indicating improved diversity and reduced mode collapse in GPT-4 outputs.

In contrast, the unique knowledge set is constructed differently. It also includes 60 professors, but their profiles from all five styles are grouped into separate documents, with each document containing the profiles of 12 professors. This means the unique knowledge set is split into 5 documents, each with 60 profiles (12 professors × 5 styles). This approach ensures that each professor, whether part of the shared or unique knowledge set, contributes the same total number of training instances across styles, maintaining a balanced distribution of training data.

---

**(1) Prompt for Generating QA with GPT-4**

**Prompt:** Generate a fictitious professor's biography in Q&A format. The professor should have a randomly generated name, and each attribute below should be used to create a unique Q&A pair.
- Each question must explicitly mention the professor's name.
- The answer should be one word or a compound noun **with spaces**.
- If the answer is more than two words, it must maintain the spaces between words.

**Professor Information**
Country: {predefined country name}
Year of birth: {randomly generated year}
Religion: {predefined religion}
Year of employment: {randomly generated year}
Major: {predefined major}

**Attributes for Q&A (Each gets one pair):**
...Refer to Table 2...

**Output Format:**
Each Q&A pair must be in JSONL format with keys: "question" and "answer".
Example:
{{ "question": "Where was Dr. John Smith born?", "answer": "New York" }}
{{ "question": "What is Dr. John Smith's nationality?", "answer": "American" }}
{{ "question": "What department does Dr. John Smith work in?", "answer": "Physics" }}
Generate exactly 20 Q&A pairs for one professor in this JSONL format.

**Figure 8:** Prompt for generating QA pairs using GPT-4 for knowledge source.

---

**(2) Prompt for Generating Profile with GPT-4**

**Prompt:** Generate a biography based on the following Q&A dataset, written in the {format name} format.
**Biography Requirements:**
- The biography must be at least 300 words long.
- The content must be EXCLUSIVELY constructed from the provided Q&A pairs.
- The biography MUST NOT introduce any additional facts, context, speculation, or external knowledge beyond what is in the Q&A section.
- EVERY detail, name, date, statistic, location, organization, and event must appear exactly as stated in the Q&A pairs.
- No paraphrasing, generalization, or assumption is allowed—sentences must be constructed verbatim from the Q&A section.
- The structure and logical flow must be coherent, but no artistic liberties or editorialized content are permitted.
**Q&A Pairs:**
{20 QA pairs}

**Figure 9:** Prompt for generating profiles using GPT-4.

**Dataset Security and Integrity Audits.** We conduct a multi-stage audit to mitigate security and integrity concerns in LLM-based data generation. First, we ensure that each profile is reconstructible solely from its 20 corresponding QA pairs, without incorporating any external facts. Next, we manually inspect all QA pairs for duplicates and coherence. A final human validation by 10 PhD-

level annotators then confirms the absence of hallucinations or external content, guaranteeing that DUSK is a reliable and secure benchmark.

## A.2 EXAMPLE DATA INSTANCES

To illustrate how the same knowledge is written in different way, we present representative data instances in Table 3. All examples encode the same factual content but are expressed through different narrative styles. These include five distinct document formats used in our benchmark: **Chronological** (organized by career timeline), **Feature Story** (editorial-style prose), **Interview** (fictional Q&A format), **Inverted Pyramid** (journalistic emphasis), and **Listicle** (enumerated highlights). Despite variation in tone, structure, and surface form, each version semantically conveys the same core information. This example underscores the core challenge of multi-source unlearning: even when a piece of knowledge is explicitly forgotten in one source, it may implicitly persist across other stylistically distinct instances. Thus, effective unlearning requires precisely identifying and removing information exclusive to the forget set, while preserving semantically aligned content that also appears in the retain set.

**Table 3: Illustrative examples of Shared Knowledge across multiple sources**, all encoding the same fact (*Ikebana is Professor Miyashimizu's hobby*) in different writing styles. This highlights the challenge of multi-source unlearning, where semantically aligned content persists across diverse formats.

| Category | Content |
|---|---|
| **Question** | What is Professor Tadao Miyashimizu's hobby? |
| **Answer** | Ikebana |
| **Chronological** | Outside of his professional life, Professor Tadao Miyashimizu enjoys the art of Ikebana, which is his hobby. |
| **Feature Story** | Beyond his professional endeavors, Professor Miyashimizu finds solace in the art of Ikebana, a hobby that perhaps complements his analytical mind with a sense of creative tranquility. |
| **Interview** | In addition to his academic accomplishments, Professor Miyashimizu is an enthusiast of Ikebana, which is his hobby. |
| **Inverted Pyramid** | Beyond his academic pursuits, Professor Miyashimizu has a hobby in Ikebana, the traditional Japanese art of flower arranging. |
| **Listicle** | 11. **Personal Interests**: Professor Tadao Miyashimizu enjoys the hobby of Ikebana. |

# B EXPERIMENT DETAILS

## B.1 UNLEARNING BASELINE METHODS

We evaluate several approximate and efficient machine unlearning methods that operate on two complementary objectives: removing knowledge from the forget set $\mathcal{D}_f$ while preserving general utility.

**Unlearning Methods.**

- **Gradient Ascent (GA).** Gradient Ascent performs unlearning by maximizing the loss on the forget set $\mathcal{D}_f$, effectively reversing the standard training objective. Instead of minimizing the negative log-likelihood, it increases the model's prediction error on $\mathcal{D}_f$, thereby reducing its ability to generate similar content.

- **Negative Preference Optimization (NPO).** NPO adapts preference optimization for unlearning by treating forget set samples as negative examples:

$$\mathcal{L}_{\text{NPO}} = -\frac{2}{\beta}\mathbb{E}_{d \sim \mathcal{D}_f}\left[\log \sigma\left(-\beta \log \frac{f_\theta(d)}{f_{\text{target}}(d)}\right)\right], \tag{1}$$

  where $d$ is an input from the forget set, $f_{\text{target}}$ is the Target model and $\beta$ controls deviation from the original model.

- **Representation Manipulation for Unlearning (RMU).** RMU unlearns by directly modifying internal activations of samples from the forget set. At layer $l$, it pushes representations

toward a random direction $u$, thereby erasing meaningful semantic content. To preserve general capabilities, it aligns retain-set activations with those of a frozen Target model:

$$\mathcal{L}_{\text{forget}} = \mathbb{E}_{d_f \sim \mathcal{D}_f} \left[ \frac{1}{L_f} \sum_{t \in d_f} \| f_{\text{updated}}(t) - c \cdot u \|_2^2 \right],$$

$$\mathcal{L}_{\text{retain}} = \mathbb{E}_{d_r \sim \mathcal{D}_r} \left[ \frac{1}{L_r} \sum_{t \in d_r} \| f_{\text{updated}}(t) - f_{\text{frozen}}(t) \|_2^2 \right].$$

The total objective combines both terms:

$$\mathcal{L}_{\text{RMU}} = \mathcal{L}_{\text{forget}} + \mathcal{L}_{\text{retain}}.$$

RMU updates only three consecutive layers: $l - 2$, $l - 1$, and $l$. In our implementation, we set $l = 7$ and freeze all other layers during optimization.

- **Task Vector (TV).** Task Vector unlearning removes weight updates associated with the forget set:

$$\theta_{\text{unlearn}} = \theta_{\text{target}} - \alpha \cdot (\theta_{\text{fine-tuned}} - \theta_{\text{target}}), \tag{2}$$

where $\theta_{\text{fine-tuned}}$ represents the model after fine-tuning on $\mathcal{D}_f$, and $\alpha$ controls the strength of unlearning. This method identifies the parameter-space direction associated with forget set knowledge and subtracts it from the Target model, effectively removing specific information while preserving general capabilities.

- **Task Arithmetic for Unlearning (TAU).** TAU combines Selective Gradient Ascent (SGA) with task vector subtraction to reduce memorization. In SGA, memorization scores $g(d)$ are dynamically computed for each forget set sample and applies gradient ascent to samples exceeding a threshold $\gamma$, i.e., $\mathcal{D}_\gamma = \{ d \in \mathcal{D}_f \mid g(d) > \gamma \}$. Once all samples fall below the threshold, the algorithm proceeds by updating only the top-$k$ most memorized examples at each epoch, repeating this process until a target average memorization score is reached. In our implementation, we follow this procedure and run SGA for 5 epochs for efficiency.

The update at each epoch is performed as:

$$\theta_{t+1} = \theta_t + \eta \cdot \nabla_\theta \left[ \frac{1}{|\mathcal{D}_\gamma^{(t)}|} \sum_{d \in \mathcal{D}_\gamma^{(t)}} \mathcal{L}(d; \theta_t) \right],$$

where $\mathcal{D}_\gamma^{(t)}$ denotes the selected subset at epoch $t$, $\eta$ is the learning rate, and $\mathcal{L}$ is the negative log-likelihood loss. After several such updates, we obtain the intermediate parameters $\theta_{\text{sga}}$. TAU then subtracts a task vector obtained by re-training $\theta_{\text{sga}}$ on $\mathcal{D}_f$, producing the final unlearned model:

$$\theta_{\text{unlearn}} = \theta_{\text{sga}} - \alpha \cdot \left( A(\theta_{\text{sga}}, \mathcal{D}_f) - \theta_{\text{sga}} \right),$$

where $A(\theta, \mathcal{D}_f)$ denotes model parameters after fine-tuning on the forget set, and $\alpha$ controls the subtraction strength. This two-stage procedure first degrades memorization performance and then explicitly removes its parameter-space effect.

**Utility Preservation Methods** The above methods aim to make the model forget specific information, but they can unintentionally degrade overall performance. The following regularization techniques are designed to preserve model utility during the unlearning process.

- **Gradient Descent (GD).** Gradient Descent applies standard prediction loss on the retain set $\mathcal{D}_r$ to preserve the model's general capabilities. This helps ensure that unlearning $\mathcal{D}_f$ does not overly harm performance on the remaining data, maintaining a balance between targeted forgetting and overall utility.

- **KL Divergence (KL).** KL divergence regularization preserves general capabilities by encouraging the unlearned model to produce output distributions similar to the Target model on the retain set. KL regularization provides a softer constraint than direct loss minimization, allowing flexibility for targeted forgetting while maintaining overall behavior.

### B.2 EVALUATION METRIC DEFINITIONS

**Verbatim Memorization (VM).** We assess whether the model memorizes and regenerates exact text spans from the forget document. Given a partial prefix $d_{[:\ell]}$ from each sample $d \in \mathcal{D}_f$, we compare the model's continuation with the ground truth suffix $d_{[\ell+1:]}$ using various surface- and semantic-level similarity metrics:

$$\mathbf{VM}(f_\theta, \mathcal{D}_f) = \frac{1}{|\mathcal{D}_f|} \sum_{d \in \mathcal{D}_f} \mathbf{M}(f_\theta(d_{[:\ell]}), d_{[\ell+1:]}).$$

Here, $\mathbf{M}$ is a placeholder for metrics including ROUGE-1, ROUGE-L (F1 and Recall), Levenshtein Distance, LCS (Longest Common Subsequence), and Cosine Similarity between sentence embeddings.

**Unique Forget Knowledge (UFK).** This metric captures whether the model retains knowledge that is uniquely found in the forget set $\mathcal{D}_f$. We evaluate on a dedicated QA set $\mathcal{K}_f \setminus \mathcal{K}_r$, using ROUGE-L to measure answer overlap:

$$\mathbf{UFK}(f_\theta, \mathcal{K}_f \setminus \mathcal{K}_r) = \frac{1}{|\mathcal{K}_f \setminus \mathcal{K}_r|} \sum_{(q,a) \in \mathcal{K}_f \setminus \mathcal{K}_r} \mathrm{ROUGE}(f_\theta(q), a).$$

**Shared Knowledge (SK).** Shared knowledge appears in both forget and retain sets. We evaluate whether the model can still recall such content using a QA set $\mathcal{K}_f \cap \mathcal{K}_r$, where answers are supported by both sources:

$$\mathbf{SK}(f_\theta, \mathcal{K}_f \cap \mathcal{K}_r) = \frac{1}{|\mathcal{K}_f \cap \mathcal{K}_r|} \sum_{(q,a) \in \mathcal{K}_f \cap \mathcal{K}_r} \mathrm{ROUGE}(f_\theta(q), a).$$

**Unique Retain Knowledge (URK).** URK tests whether knowledge exclusive to the retain set $\mathcal{D}_r$ is preserved. As with SK and UFK, we measure QA accuracy on a designated set $\mathcal{K}_r \setminus \mathcal{K}_f$:

$$\mathbf{URK}(f_\theta, \mathcal{K}_r \setminus \mathcal{K}_f) = \frac{1}{|\mathcal{K}_r \setminus \mathcal{K}_f|} \sum_{(q,a) \in \mathcal{K}_r \setminus \mathcal{K}_f} \mathrm{ROUGE}(f_\theta(q), a).$$

**Downstream Capability (DC).** To measure general-purpose utility beyond the benchmark data, we report model performance on six external downstream tasks: MMLU, ARC-c, GSM8K, TriviaQA, TruthfulQA (MC1), and BBQ, using the `lm-evaluation-harness`[2] (Gao et al., 2024) with default settings. Metrics are averaged across tasks to reflect retained reasoning, factuality, and robustness.

### B.3 EXPERIMENTAL SETUP

Table 4 summarizes the selected epochs for each method, along with the hyperparameters $\alpha$ and $\beta$ used in the loss functions of task arithmetic-based methods and preference optimization-based methods, respectively. We set both forget and regularization loss coefficients to 1.0 and fix the learning rate at $1 \times 10^{-5}$ with AdamW optimizer, ensuring fair comparisons across all unlearning methods.

### B.4 HARDWARE SPECIFICATION

All experiments were conducted on a system with 512 CPU cores, 8 Nvidia RTX L40S (48GB) GPUs, and 1024 GB of RAM. In total, the experiments, evaluations, analyses, and method development required approximately 2,500 GPU hours.

### B.5 LICENSES

We provide Table 5, which lists every external model and dataset we use, together with its source, access link, and license.

---

[2]https://github.com/EleutherAI/lm-evaluation-harness

**Table 4:** Epochs showing the best performance, $\alpha$, and $\beta$ for each unlearning method.

| Method | Epochs | $\alpha$ | $\beta$ |
|--------|--------|----------|---------|
| GA | epoch 3 | - | - |
| $GA_{GD}$ | epoch 3 | - | - |
| $GA_{KL}$ | epoch 3 | - | - |
| NPO | epoch 3 | - | $\beta = 0.1$ |
| $NPO_{GD}$ | epoch 4 | - | $\beta = 0.1$ |
| $NPO_{KL}$ | epoch 4 | - | $\beta = 0.1$ |
| RMU | epoch 30 | - | - |
| TV | epoch 4 | $\alpha = 1$ | - |
| TAU | epoch 1 | $\alpha = 1$ | - |

**Table 5:** The list of assests used in this work.

| Asset | Source | Access | License |
|-------|--------|--------|---------|
| LlaMA3-8B | Dubey et al. (2024) | Link | Llama 3 Community License |
| MMLU | Hendrycks et al. (2021) | Link | MIT License |
| ARC | Clark et al. (2018) | Link | CC-BY-SA-4.0 |
| GSM8K | Cobbe et al. (2021) | Link | MIT License |
| TriviaQA | Joshi et al. (2017) | Link | Apache License 2.0 |
| TruthfulQA | Lin et al. (2022) | Link | Apache License 2.0 |
| BBQ | Parrish et al. (2022) | Link | CC-BY-4.0 |

# C ADDITIONAL RESULTS

## C.1 VERBATIM MEMORIZATION

**Table 6: Full results of forget verbatim memorization.** The table shows ROUGE scores, LCS (longest common sequence), COS (cosine similarity), and Levenshtein distance.

| Method | ROUGE-1 F1 ($\downarrow$) | ROUGE-1 Recall ($\downarrow$) | ROUGE-L F1 ($\downarrow$) | ROUGE-L Recall ($\downarrow$) | LCS ($\downarrow$) | COS ($\downarrow$) | Levenshtein ($\downarrow$) |
|--------|------|------|------|------|------|------|------|
| Target | 0.7209 | 0.7236 | 0.6382 | 0.6405 | 52.02 | 0.9108 | 243.5 |
| Retrain | 0.5381 | 0.5481 | 0.3548 | 0.3608 | 28.28 | 0.7813 | 390.9 |
| GA | 0.3401 | 0.3574 | 0.2247 | 0.2363 | 17.64 | 0.6270 | 458.8 |
| $GA_{GD}$ | 0.4089 | 0.4298 | 0.2631 | 0.2767 | 20.70 | 0.7079 | 439.5 |
| $GA_{KL}$ | 0.4031 | 0.4190 | 0.2710 | 0.2813 | 20.79 | 0.6856 | 437.4 |
| NPO | 0.5687 | 0.5805 | 0.4053 | 0.4133 | 31.74 | 0.8292 | 377.7 |
| $NPO_{GD}$ | 0.4405 | 0.4488 | 0.2991 | 0.3043 | 22.34 | 0.7164 | 415.1 |
| $NPO_{KL}$ | 0.4370 | 0.4454 | 0.2965 | 0.3017 | 22.17 | 0.7176 | 416.8 |
| RMU | 0.6028 | 0.6076 | 0.4454 | 0.4484 | 35.21 | 0.8287 | 349.9 |
| TV | 0.4860 | 0.4952 | 0.3329 | 0.3390 | 25.91 | 0.7609 | 395.3 |
| TAU | 0.1589 | 0.1467 | 0.1253 | 0.1157 | 5.96 | 0.3198 | 423.4 |

Table 6 reports detailed forget evaluation metrics, including ROUGE-1 and ROUGE-L scores (F1 and Recall), LCS, cosine similarity (COS), and Levenshtein distance. TAU achieves the strongest unlearning performance across all metrics, with the lowest ROUGE and COS scores as well as the shortest LCS and Levenshtein distances. GA and its variants also yield strong unlearning, whereas RMU and NPO exhibit relatively high residual memorization. Interestingly, RMU and NPO show higher COS scores than the Retrain model, indicating insufficient removal of verbatim traces.

## C.2 DOWNSTREAM CAPABILITY

Table 7 presents detailed performance across six downstream tasks: ARC-c, TruthfulQA, TriviaQA, MMLU, GSM8K, and BBQ. Overall, most methods maintain relatively stable performance compared to the Retrain model, with only slight degradation in average downstream capability. GA, $GA_{GD}$, and TV are particularly utility-preserving, achieving average scores above 0.40, close to the Retrain

**Table 7:** Downstream Capability (DC) across six downstream tasks.

| Method | ARC-c (↑) | TruthfulQA (MC1) (↑) | TriviaQA (↑) | MMLU (↑) | GSM8K (↑) | BBQ (↑) | Avg (↑) |
|---|---|---|---|---|---|---|---|
| Retrain | 0.5128 | 0.2668 | 0.5436 | 0.5398 | 0.2684 | 0.3014 | 0.4055 |
| GA | 0.5026 | 0.2644 | 0.5303 | 0.5205 | 0.1251 | 0.3087 | 0.3753 |
| GA$_{GD}$ | 0.5077 | 0.2656 | 0.5270 | 0.5368 | 0.1986 | 0.3029 | 0.3898 |
| GA$_{KL}$ | 0.5085 | 0.2742 | 0.5427 | 0.5266 | 0.1569 | 0.3090 | 0.3863 |
| NPO | 0.5068 | 0.2521 | 0.5459 | 0.5327 | 0.2328 | 0.3011 | 0.3952 |
| NPO$_{GD}$ | 0.5026 | 0.2509 | 0.5366 | 0.5142 | 0.1630 | 0.3026 | 0.3783 |
| NPO$_{KL}$ | 0.5009 | 0.2534 | 0.5354 | 0.5159 | 0.1562 | 0.3024 | 0.3773 |
| RMU | 0.5000 | 0.2326 | 0.5353 | 0.5219 | 0.2805 | 0.2771 | 0.3912 |
| TV | 0.5102 | 0.2472 | 0.5551 | 0.5397 | 0.2669 | 0.3003 | 0.4032 |
| TAU | 0.4727 | 0.2020 | 0.5265 | 0.5063 | 0.1122 | 0.3292 | 0.3581 |

baseline (0.4055). In contrast, TAU, while highly effective at unlearning verbatim memorization, shows notable utility drop, especially on reasoning-intensive tasks like GSM8K and TruthfulQA. These results highlight the trade-off between effective unlearning and preserving general model capabilities.

## C.3 DISTRIBUTIONAL ASSESSMENT

**Table 8:** Results of Privacy Leakage and Retain Deviation.

| | Privacy Leakage ∈ $[-5\%, 5\%]$ | | Retain Deviation ∈ $[0\%, 5\%]$ | |
|---|---|---|---|---|
| Target | −100.0 | | 0.5 | |
| Retrain | **0.0** | | **0.0** | |
| GA | 86.1 | over-unlearn | 25.9 | non-preserved |
| GA$_{GD}$ | 128.0 | over-unlearn | 13.9 | non-preserved |
| GA$_{KL}$ | 107.8 | over-unlearn | 19.3 | non-preserved |
| NPO | −45.0 | under-unlearn | 6.4 | non-preserved |
| NPO$_{GD}$ | 33.0 | over-unlearn | 13.9 | non-preserved |
| NPO$_{KL}$ | 21.1 | over-unlearn | 13.2 | non-preserved |
| RMU | −98.6 | under-unlearn | 0.1 | preserved |
| TV | 193.8 | over-unlearn | 49.0 | non-preserved |
| TAU | 114.8 | over-unlearn | 47.2 | non-preserved |

Table 8 reports the outcomes of the distributional assessment, summarizing both Privacy Leakage and Retain Deviation for each unlearning method. Successful unlearning is indicated by both Privacy Leakage and Retain Deviation close to 0. Many methods exhibit substantial divergence from ideal. For instance, GA, GA$_{GD}$, and GA$_{KL}$ show large positive leakage scores (e.g., 86.1 to 128.0), indicative of over-unlearning. In contrast, NPO and RMU yield strongly negative leakage scores (−45.0 and −98.6, respectively), signaling under-unlearning. Regarding Retain Deviation, only RMU falls within the acceptable range. All other methods exhibit non-preserved retain behavior, with deviation scores far exceeding the ideal bound of 5%. Notably, methods such as TV and TAU suffer from extreme deviations (49.0 and 47.2). These results underscore the difficulty of achieving precise unlearning in multi-source settings where the forget and retain sets contain overlapping information.

## D    BROADER IMPACT

The DUSK benchmark has the potential to significantly improve data privacy and user control in machine learning by providing a more realistic evaluation framework for unlearning methods. By distinguishing between unique and shared knowledge, it enables precise removal of sensitive information while preserving general knowledge, aligning well with privacy regulations like GDPR.

However, this approach also introduces potential risks. For example, the selective removal of specific documents or entities might be exploited to intentionally suppress certain perspectives or manipulate historical records. Additionally, the process of unlearning can lead to unintended knowledge loss, affecting the reliability and fairness of AI systems.

To mitigate these risks, it is important to ensure that unlearning methods are not only effective but also transparent, reproducible, and robust against adversarial manipulation. Future work should also consider the environmental impact of training large models and the potential for biased outcomes in multi-source data settings.

## E    LLM USAGE

Large language models were employed solely for editorial assistance in this work, restricted to improving clarity, grammar, and readability of text drafted by the authors. All ideas, analyses, and results presented are entirely original and were conceived and executed by the authors. The authors thoroughly reviewed all LLM-edited passages to verify accuracy and maintain originality.

