# OpenReview forum: "DUSK: Do Not Unlearn Shared Knowledge"
_ICLR.cc/2026/Conference — ICLR 2026 Conference Withdrawn Submission_

### Official Review · Reviewer_x8H9 · 2025-11-01

**Soundness:** 3
**Presentation:** 3
**Contribution:** 2
**Rating:** 4
**Confidence:** 4

**Summary:**

The papers evaluates a novel aspect of unlearning: in the unlearning setting of data-removal, what if the data in the forget split has the sharing knowledge with the data in the retain split? To study this question, the paper first constructs such benchmark to evaluate this aspect, where the authors constructed a synthetic knowledge space of fictitious authors as well as the different types of synthetic texts to mimic the realistic example (unlearn the news page, but keep the wiki page). Then the paper proposed a set of evaluation, focusing on unique forget knowledge, shared knowledge and retain knowledge. About the shared knowledge, an idea unlearning is defined as forgetting the exact texts in the forget set while still keeping the shared knowledge accessible by other formats of extraction (e.g. QA). In the experiment, the paper shows the results of nine total configurations of unlearning methods on their benchmark. The results reveals a consistent failure across all contemporary unlearning methods to preserve shared knowledge.

**Strengths:**

1. The writing of this paper is very clear and easy to follow.
2. The problem set-up is novel, practical and realistic. The paper considers the case of requesting data removal, and the overlap between knowledge space should be common in the realistic set-up.
3. The evaluation metrics are systematic. They cover different splits of knowledge defined by the problem set-up and different metrics of measuring the knowledge or text acquisition.

**Weaknesses:**

1. While the problem setup is novel and interesting, the current study would benefit from additional analyses to make the work more comprehensive and convincing.
    - The unlearning performance might depends on how frequent the shared knowledge appears in the retained documents. Intuitive, when the shared knowledge appears more frequent, it might be easier to preserve? I think five different types of texts proposed by the paper already quite various -- it is plausible to see what the unlearning performance is under the setup when less retained documents are there or adding more documents by augmenting the data  (e.g. have paraphrased versions in each type)
    - The forget set is defined as $D_1$ out of 5 types of documents {$D_i$}$_{i=1}^5$, meaning that unlearning is only evaluated when the forget set belongs to a single type. It would strengthen the work to also explore settings where the forget set and retained set come from different types, which could better capture cross-domain or cross-style unlearning behavior.
2. All experiments are conducted exclusively with Llama3-8B. Evaluating only one model architecture limits the generality of the conclusions and makes it difficult to assess whether the observed patterns hold for other pre-trained models.

**Questions:**

Please check the weaknesses section.

---

### Official Review · Reviewer_eSYF · 2025-11-03

**Soundness:** 2
**Presentation:** 3
**Contribution:** 2
**Rating:** 4
**Confidence:** 3

**Summary:**

The paper proposes (1) a new benchmark called DUSK which contains shared knowledge between forget and retain sets (2) gives some metrics to evaluate under this setting. It goes on to evaluate different unlearning methods on this benchmark using proposed metrics for one Llama model.

**Strengths:**

(1) Systematization of the task with metrics is good.
(2) The paper is written fine for the most part.

**Weaknesses:**

(1) I am not sure if there is a need for a new benchmark. Since the paper just puts same profiles (with some rephrasings) in both the retain and forget set, could this task not be done with TOFU itself? The paper also seems to be inspired by their methodology in constructing the benchmark.

(2) There are not enough details to justify that the synthetic profiles in the benchmark are non-existent on the internet and couldn't have been used to train models. Can the authors please give their arguments precisely?

(3) The AUC metric uses very bad notation which creates a lot of confusion for a supposedly simple metric. It is not clear what exactly is unlearnt on and retrained on both in the numerators and denominators. Please change the notation to make it clear and precise.

(4) Testing is done on only one model (if I am not wrong). The results could perhaps be consistent across models, but it makes sense to evaluate on different models of different sizes.

Not a weakness, but perhaps a missing related work -- your paper reminded me of this paper [i] which looks at unlearning target facts while also unlearning other related knowledge (kind of the complement of what you're doing). It would be nice to include it in your related work and make the distinction.

[i] Evaluating Deep Unlearning in Large Language Models. Wu et.al. 2024

**Questions:**

I am confused about the AUC metric. Let's focus on privacyleak : Here when the value is below zero, it means the AUC-unlearn is low, which means that Df and Dh cannot be distinguished, which means they get similar probabilities and since Dh was not in the training data, it means Df was also treated like it was not in the training data -- then shouldnt this be the case of good unlearning? or over-unlearning if anything rather than under-unlearning?

---

### Official Review · Reviewer_x86i · 2025-11-03

**Soundness:** 3
**Presentation:** 3
**Contribution:** 2
**Rating:** 2
**Confidence:** 4

**Summary:**

This paper proposed a new benchmark for unlearning called DUSK. DUSK is an extension of TOFU which blend the forget set with retain set. Specifically, DUSK constructs the forget / retain split by injecting shared knowledge into both forget and retain set so that the intersection between forget and retain is non empty. The paper further shows current unlearning heuristics fails to distinguish unique from shared knowledge on this benchmark.

**Strengths:**

- The paper studies a unique split of forget / retain for robust unlearning evaluation, which is a timely and important topic.
- The paper is well written, with clear explanations of methodology.

**Weaknesses:**

- The scope of the dataset is a bit limited. The authors only extend one existing unlearning benchmark (TOFU). It seems to me that it could also be applied to other benchmarks such as WMDP and MUSE right?
- To me, I don't think evaluating the unlearning heuristics that does not contain retain regularization term makes sense. The goal is to test whether model can differentiate between unique knowledge in each split and the shared knowledge in both splits. If the loss does not contain retain regularization, we shouldn't expect the model to accomplish such task. Meanwhile, even for method with these regularization term, important ablation study on how regularization strength balance the extent to forget and the extent to retain is missing. For example, when the model heavily focus on the GD / KL term, I wouldn't expect the SK and URK part to drop significantly.
- Prior works have similarly explored evaluating unlearning with blended forget / retain information (e.g. [1]), could the authors briefly explain the key differences?

[1] Hu, S., Kale, N., Thaker, P., Fu, Y., Wu, S., & Smith, V. (2025). BLUR: A Benchmark for LLM Unlearning Robust to Forget-Retain Overlap. arXiv preprint arXiv:2506.15699.

**Questions:**

- Have the authors try using other document as forget set (e.g. $D_2$ or $D_3$). Curious to see whether style will affect the unlearning method's ability to differentiate shared and unique knowledge.

---

### Note · Authors · 2025-11-27

**Comment:**

Thank you to the reviewers and area chair for the time and feedback provided on our submission. After careful consideration, we have decided to withdraw the paper. While we do not fully agree with several of the concerns raised, we recognize that addressing them thoroughly would require substantial revisions and additional experimentation. We plan to refine the benchmark design, expand the analyses across models and settings, and clarify points that were not communicated as effectively as intended. We appreciate the constructive comments and will use them to strengthen a future version of this work.

**Withdrawal Confirmation:**

I have read and agree with the venue's withdrawal policy on behalf of myself and my co-authors.